# Physician payment models and cardiac imaging in patients at low cardiovascular risk: A population-based cohort study in Alberta, Canada

Yewande Kofoworola Ogundeji[1], Amity E. Quinn[1,2], Derek S. Chew[1,2,3,4], Flora Au[2], Stephen B. Wilton[1,2,3], Matthew T. James[1,2,3,4], Marcello Tonelli[1,2,3,4], Braden J. Manns[1,2,3,4]*

1 Department of Community Health Sciences, Cumming School of Medicine, University of Calgary, Alberta, Canada, 2 Department of Medicine, Cumming School of Medicine, University of Calgary, Alberta, Canada, 3 Libin Cardiovascular Institute, Cumming School of Medicine, University of Calgary, Alberta, Canada, 4 O'Brien Institute for Public Health, Cumming School of Medicine, University of Calgary, Alberta, Canada

* bjmanns@ucalgary.ca

## Abstract

### Background

Many factors beyond patient need influence the care that patients receive, including the way physicians are paid, and how services are delivered. In Alberta, outpatient non-invasive cardiac imaging ("cardiac imaging") is paid for publicly but performed at private, for-profit (investor/physician owned) facilities. We investigated patient, physician, and geographic factors associated with cardiac imaging in patients at low cardiovascular risk seeing specialist physicians in Alberta, Canada.

### Methods

This was a population-based retrospective cohort study using administrative health data from Alberta, Canada, where nearly all outpatient cardiac imaging is done at privately for-profit community-based facilities. We used administrative health data to identify a cohort of adult (aged ≥18 years) patients at low cardiovascular risk who were assessed by a cardiologist or internal medicine specialist for a new out-patient visit for a cardiac-related reason between April 1, 2011 and December 30, 2019 in Alberta. The primary outcome was cardiac imaging. Explanatory variables included patient and physician characteristics, including payment model (fee for service (FFS) or salary-based), and geography. We used multilevel, multivariable logistic regression models to measure the association between these factors and cardiac imaging.

**Data availability statement:** We cannot make our dataset available to other researchers due to our contractual arrangements with the provincial health ministry (Alberta Health), who is the data custodian. (Please see response to reviewers on this point). Researchers may make requests to obtain a similar dataset upon which they could replicate these analyses at https://absporu.ca/research-services/service-application/.

**Funding:** A Canadian Institutes of Health Research Foundation Grant to Manns and a Data Collection Grant from the Clinical Research Fund, University of Calgary to Ogundeji and Manns funded this study. The funders had no role in study design, data collection and analysis, decision to publish, or preparation of the manuscript.

**Competing interests:** The authors have declared that no competing interests exist.

## Results

We identified 398,095 patients at low cardiovascular risk, of whom 27.5% received at least one cardiac imaging test. Compared to those seen by FFS cardiologists (and controlling for patient and geographic differences), patients seen by salary-based internal medicine specialists had the lowest odds of receiving cardiac imaging (OR=0.055, P<0.001, CI 0.036–0.086), followed by those seen by FFS internal medicine specialists (OR=0.010, P<0.001, CI 0.068–0.14), and salary-based cardiologists (OR=0.27, P<0.001, CI 0.16–0.45). Findings were robust across multiple sensitivity analyses.

## Conclusions

Physician payment models and specialty are strongly associated with non-invasive cardiac imaging among patients at low cardiovascular risk.

## Introduction

There are many factors beyond patient need that influence care delivery, including the way physicians are paid, and how services are delivered. Past studies have noted significant variation in the use of non-invasive cardiac imaging (hereafter called "cardiac imaging") based on both patient and physician characteristics [1]. Use of cardiac testing in people at low cardiovascular risk can lead to false positive tests, patient anxiety, and additional invasive testing for diagnostic clarity. Overuse of cardiac testing has been highlighted by Choosing Wisely Canada as being of concern [2,3]. Given the increasing availability of cardiac diagnostic testing offered through private for-profit facilities in many provinces [4] and both over and underutilization of cardiac diagnostic testing, it is important to ensure that patients receive only the cardiac testing they need.

Physician payment models have been shown to influence patient care, with fee-for-service (FFS) typically increasing rates of care [5,6] compared with alternate payment models, a phenomena called supplier-induced demand [7]. In health systems with for-profit cardiac testing, where FFS physicians ordering tests are also paid to interpret these imaging tests, and/or have an equity interest in an imaging facility, there may be an additional financial incentive to order such testing [1].

Past studies examining the use of low-value cardiac imaging have focused on describing rates of individual tests in primary care or included limited predictive factors [8,9]. Several physician characteristics are associated with substantial variation in primary care testing rates (e.g., prostate specific antigen or bone mineral density scans) despite similar patient complexity, including physician male gender, older age, and FFS payment [9]. However, few studies have investigated whether specialty physician practice types or payment models influence the utilization of cardiac imaging.

Our study had two aims: (i) to investigate patient, physician, and geographic-level factors associated with cardiac imaging testing rates among patients at low

cardiovascular risk seen by cardiologists and internal medicine specialists, and (ii) to describe the characteristics of specialists who care for patients with a higher frequency of cardiac imaging.

## Materials and methods

### Overview

We used administrative health data to identify a population-based retrospective cohort of patients at low cardiovascular risk who were assessed by a cardiologist or internal medicine specialist for a new cardiac-related outpatient visit. We investigated the factors associated with the use of cardiac imaging using multilevel regression models to account for variability attributable to patient, physician, and geographic-level factors. This study was approved by University of Calgary's Conjoint Health Research Ethics Board (REB16–1575), who granted a waiver of informed consent since this was considered secondary use of administrative health data as defined within Alberta's Health Information Act. All data were fully anonymized before they were accessed, starting on 23/7/2023.

### Study setting

This study was performed in Alberta, Canada, where nearly all outpatient cardiac imaging is done at private for-profit community-based facilities. Testing is fully paid for by the publicly funded Alberta Health Insurance Plan for Alberta residents, with no out of pocket costs for patients [10]. Fees paid by the Alberta Health Insurance Plan for Echocardiogram, MPI, and exercise stress test are $250.36, $248.51-$426.41, and $114.60 respectively. In addition, any type of physician is able to order cardiac imaging for any indication. Supply of cardiac imaging is high in the major urban centers in Alberta, with many private facilities and wait times that are anecdotally often less than a week for echocardiography. Medical specialists in Alberta are paid by FFS, where they receive payment for each clinical service provided (including interpretation of cardiac imaging for those accredited to do so), or by a salary-based payment model [11]. In Alberta, while physicians theoretically have the opportunity to choose or switch payment models, there are limited positions (contracts) with salary-based payment models. Salary-based payment models are traditionally more common in teaching hospitals within Alberta's two large cities. At times, physicians who might prefer salary-based payment models end up on FFS as no salary positions were available at the time they were hired [12].

### Data sources

We used population-based, administrative health data from the Interdisciplinary Chronic Disease Collaboration (ICDC) data repository, which links laboratory and administrative health data for all Albertan adults (aged ≥18 years), including patient demographics, ambulatory and inpatient records, practitioner claims, physician characteristics, and dispensed pharmaceutical drugs [13]. These datasets have been used in prior studies focusing on cardiac imaging and have complete population-based coverage of a large geographically defined area [8,14].

### Cohort creation (study population)

To identify patients who were at low cardiovascular risk ('low-risk patients'), we first identified adult patients (aged ≥18 years) seeing a cardiologist or internal medicine specialist for an initial outpatient clinic for a suspected cardiac-related reason (S1 Table) between April 1, 2011 and December 30, 2019. We then excluded people with the following manifest cardiovascular disease (CVD) or traditional cardiac risk factors: (i) a prior diagnosis of CVD (MI, CHF, stroke, peripheral vascular disease, ischemic heart disease, valvular heart disease, or coronary artery bypass graft (CABG) or percutaneous coronary intervention (PCI), (ii) traditional cardiac risk factors including the following subgroups: men aged >=50 years with diagnosed hypertension and women >=60 years with diagnosed hypertension; those with diabetes and/or chronic kidney disease (eGFR < 60 mL/min/1.73m$^2$ or A3 albuminuria), and those with dyslipidemia (LDL >=3.5 mmol/l) or those on

statin, fibrates, bile acid sequestrants, nicotinic acid and derivatives or other lipid-modifying agents (Fig 1). This definition of low-risk population aligned closely with previously published studies on patients at low cardiovascular risk [9,15,16].

We identified medical comorbidities using validated administrative codes [13] (S1 and S2 Tables). Finally, we excluded physicians seeing fewer than 10 patients at low cardiovascular risk in a given year [17], since they may not have a stable practice pattern.

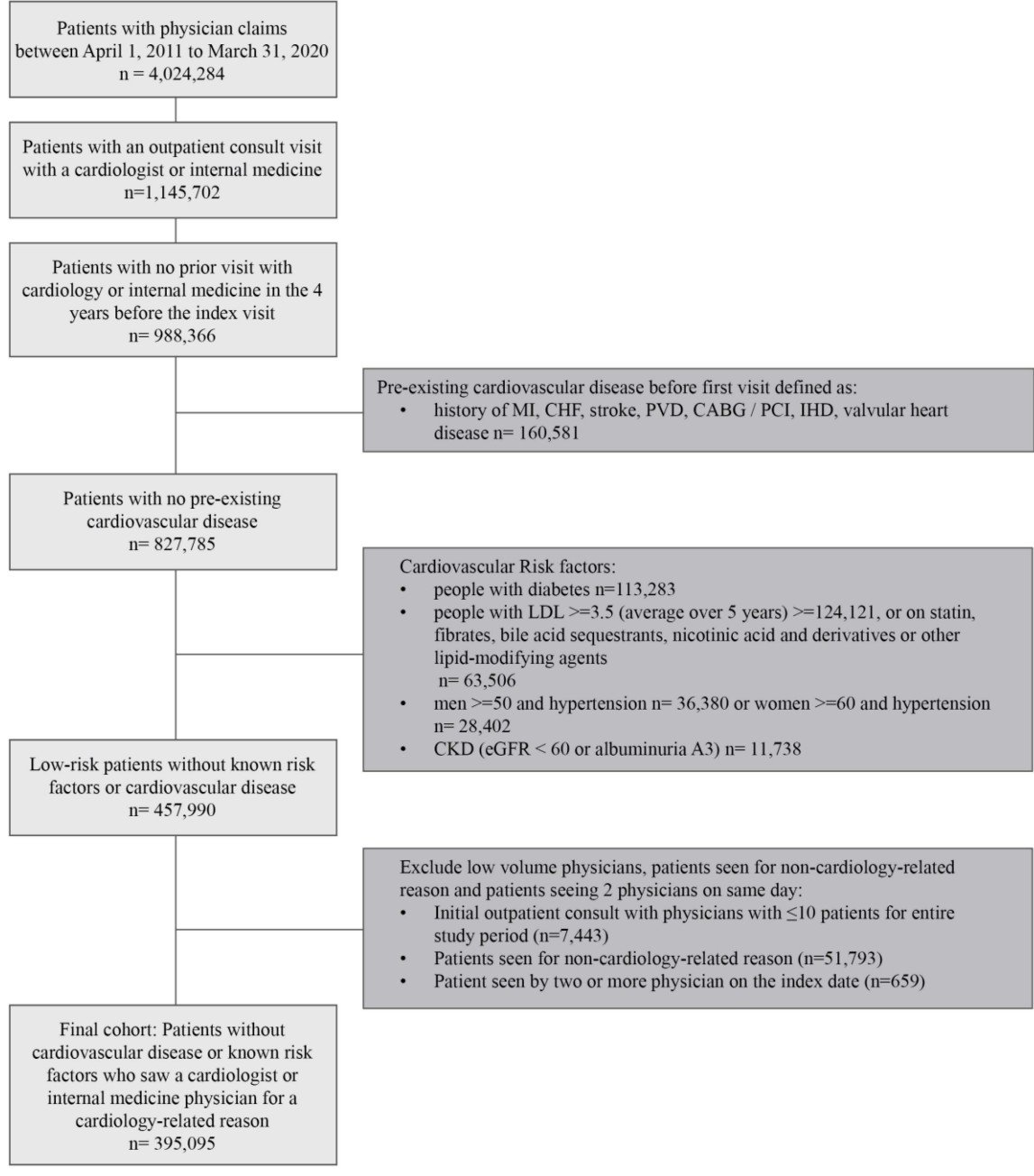

**Fig 1. Patient cohort flow diagram.**

## Outcomes

The primary outcome was a physician claim for at least one cardiac imaging test (i.e., transthoracic echocardiography, nuclear myocardial perfusion imaging, and exercise treadmill testing) occurring within the period of 30 days prior to and 90 days after the initial specialist visit (S3 Table). This time period was chosen because some specialists routinely order testing in advance of their new visits while others order them afterwards. Moreover, there is a short wait time (<30 days) for cardiac imaging studies in Alberta, but often long waits for specialist consultation.

A secondary outcome was frequency of cardiac imaging for patients seeing each physician. We categorized physicians by the proportion of low-risk patients who received cardiac imaging during the study period. Physicians with the highest tertile of cardiac imaging use were categorized as 'frequent users.'

## Covariates

We used data from the ICDC Repository [13] to collect covariates for patients at low cardiovascular risk, including demographic data, physician claims, hospital discharge abstract data, medication data, and validated algorithms to define patient comorbidities [18].

For physicians, gender, age, years in practice, physician specialty, and international medical graduate status were available in an anonymized fashion from the College of Physicians and Surgeons of Alberta. We collected information on physician payment model (FFS or salary-based), geographic zone of practice (Alberta has five health zones, including three rural zones), clinical workload, and average patient volume (mean patient visits per day on days with at least one claim) from physician claims data sets. We assessed clinical workload based on the annual percentage of days with at least one claim.

Since previous research indicated variation in Alberta specialist practice patterns by payment model [19,20], we tested the interaction between the physician payment model and specialty variables and identified a strong interaction (P<0.001). Therefore, we created four mutually exclusive physician and payment model categories in analyses: salary-based internal medicine specialists, FFS internal medicine specialists, salary-based cardiologists, and FFS cardiologists.

## Analysis

We measured baseline characteristics and compared them across physician specialty and payment models. We used a multilevel logistic regression model with random effects for physician and geographic zone to assess the association of receiving cardiac imaging with patient characteristics (age, sex, and median household income (aggregated at the patient's postal code level)), physician characteristics (age, gender, specialty, and payment model), and geographic zone. Patients with incomplete or missing data for any of the listed variables were excluded (1.8%), under the assumption that data were missing completely at random. We used multiple logistic regression to assess the odds of a physician being a frequent user based on their specialty and payment model, while controlling for other variables including physician gender, international medical graduate status, age, and years since medical school graduation. Analyses were done using Stata version 15.

## Sensitivity analysis

To address a variety of potential biases that might impact the use of cardiac imaging, we conducted sensitivity analyses. First, acknowledging that diagnostic codes in physician claims data are not always accurate, we expanded our baseline patient cohort to include any new patient visit, not limited to cardiac-related reasons, to a cardiologist or internal medicine specialist. Second, we included only nuclear myocardial perfusion imaging or exercise treadmill stress test in the outcome, given that our patient cohort was at low cardiovascular risk (and therefore would be unlikely to need imaging to assess

for coronary ischemia) but that patients might have had other signs or symptoms for structural heart disease warranting echocardiography that we could not identify from administrative data. Third, we restricted the cohort to patients with index visits for anginal or anginal-equivalent diagnostic codes and again included only nuclear myocardial perfusion imaging or exercise treadmill stress test in the outcome. Fourth, because we did not have data on the patients' presenting symptoms or physical findings, we did a fourth sensitivity analysis excluding patients who were diagnosed with cardiovascular disease, arrythmia, or required a pacemaker over the subsequent year. Finally, we repeated the baseline analyses including an additional category of physicians – FFS physicians with accreditation in cardiac testing interpretation who also bill to interpret imaging tests (FFS interpreter) – splitting the FFS cardiologist category into those who do and do not interpret imaging tests. We defined FFS interpreter as physicians with >100 imaging claims identified across their total patient population in physician claims data during the study period [1].

## Results

### Population and demographic characteristics

We included 398,095 low-risk patients (mean age, 42±SD 13.7 years) (Table 1) and 8.8%, 75.2%, 1.4%, and 14.6% were seen by salary-based internal medicine specialists, FFS internal medicine specialists, salary-based cardiologists, and FFS cardiologists, respectively. Most (91.0%, n = 362,064) lived in an urban location (Table 1). Our patient cohort was seen by 699 physicians. Among physicians, the mean age was 43.5±SD 11.2 years, 69.0% (n = 482) were male, and 65.5% (n = 458) were graduates of Canadian medical schools. Most physicians were internal medicine specialists (78.8%, n = 551), and reimbursed by FFS (76.1%, n = 532) (Table 1).

### Patient and physician characteristics associated with cardiac imaging in low-risk patients

Overall, 27.5% of our cohort received at least one cardiac imaging test. Of those seeing salary-based internal medicine specialists, FFS internal medicine specialists, salary-based cardiologists, and FFS cardiologists, 4.4%, 21.4%, 18.6%, and 73.3%, had cardiac testing, respectively (S4 Table). While patients seeing FFS cardiologists made up only 14.6% of the patient cohort, these patients represented about 39.0% of those who received at least one cardiac imaging test.

Echocardiography, exercise treadmill testing, and nuclear myocardial perfusion imaging were ordered 69,740, 66,103, and 10,266 times, respectively (Fig 2). As expected, more cardiac testing was ordered in older patients. Male patients were more likely to undergo testing, compared to female patients, and there was significant variation by zone, particularly for myocardial perfusion imaging which was more common in urban zones (Fig 2).

Fig 3 summarizes the results of the multilevel logistic regression exploring the association between patient and physician characteristics and use of cardiac imaging in low-risk patients. Patients who were older, male, and had a higher household income were more likely to receive cardiac imaging though the coefficients were small. After controlling for patient and geographic-level differences, patients seen by salary-based internal medicine specialists, FFS internal medicine specialists, and salary-based cardiologists were significantly less likely to receive cardiac imaging compared to patients seeing FFS cardiologists [OR=0.055, 95% CI 0.036–0.086; OR=0.010, 95% CI 0.068–0.14; OR=0.27, 95% CI 0.16–0.45] (Fig 3).

When we explored the proportion of variation attributable to patient, physician, and geographic factors, we found that most of the variation (54.0%) was at the physician level (S5 Table). Even after controlling for all measured fixed effects, there was still 30.6% unexplained variation at the physician level and 15.9% at the geographic zone level.

To enhance the real-world understanding of these results, we used coefficients from the model used to generate Fig 3 to estimate the likelihood of cardiac testing for patients seeing the four different types of physicians in different scenarios, identifying substantial differences (S6 Table). For instance, for a female patient, aged 41–60, with median income and fewer than three non-cardiac comorbidities, seeing a 50-year-old male physician in an urban zone, the likelihood of cardiac imaging varied between 7.8% (for a salary-based internal medicine specialist) to 60.4% (for a FFS cardiologist).

**Table 1. Demographic characteristics of patients at low cardiovascular risk and their specialist physicians.**

**Patient Characteristics**

| | Overall (n = 398,095) | Patients seeing salary-based specialist physicians (n = 40,524) | | Patients seeing FFS[+] specialist physicians (n = 357,571) | |
|---|---|---|---|---|---|
| | | Internal medicine n = 35,141 | Cardiologist n = 5,383 | Internal medicine = 299,402 | Cardiologist n = 58,169 |
| Patient age, mean (SD), years | 42.0 (13.7) | 39.4 (13.5) | 41.6 (15.2) | 41.7 (13.6) | 45.3 (13.7) |
| **Patient age, % (n), years** | | | | | |
| <=40 | 47.3 (188,162) | 58.8 (20,658) | 46.9 (2,527) | 48.3 (144,670) | 34.9 (20,307) |
| 41-60 | 42.1 (167,479) | 32.9 (11,546) | 37.3 (2,009) | 41.6 (124,481) | 50.6 (29,443) |
| 61-80 | 9.4 (37,513) | 7.3 (2,554) | 10.9 (589) | 9.0 (27,012) | 12.7 (7,358) |
| >81 | 0.6 (2,347) | 0.8 (266) | 0.8 (42) | 0.5 (1,571) | 0.8 (468) |
| missing | 0.6 (2,594) | 0.2 (117) | 4.1 (216) | 0.6 (1,668) | 1.0 (593) |
| **Patient sex, % (n)** | | | | | |
| Female | 56.9 (226,372) | 70.6 (24,815) | 49.7 (2,677) | 56.6 (169,318) | 50.8 (29,562) |
| Male | 42.5 (169,128) | 29.1 (10,209) | 46.3 (2,490) | 42.9 (128,415) | 48.2 (28,014) |
| missing | 0.6 (2,595) | 0.3 (117) | 4.0 (216) | 0.5 (1,669) | 1.0 (593) |
| **Comorbidities*, % (n)** | | | | | |
| 0 comorbidity | 51.2 (203,915) | 46.3 (16,264) | 51.6 (2,778) | 51.1 (152,985) | 54.8 (31,888) |
| 1 comorbidity | 30.8 (122,513) | 33.3 (11,698) | 30.9 (1,661) | 30.6 (91,642) | 30.1 (17,512) |
| 2 comorbidities | 12.0 (47,729) | 13.3 (4,674) | 11.8 (635) | 12.1 (36,296) | 10.5 (6,124) |
| 3 or 4 comorbidities | 5.4 (21,591) | 6.4 (2,240) | 5.3 (288) | 5.6 (16,616) | 4.2 (2,447) |
| 5 or more comorbidities | 0.6 (2,347) | 0.7 (265) | 0.4 (21) | 0.6 (1,863) | 0.4 (198) |
| **Neighbourhood income quintile, %(n)** | | | | | |
| Lowest | 24.0 (95,423) | 21.9 (7,686) | 16.2 (870) | 25.0 (74,893) | 20.6 (11,974) |
| 2nd | 20.6 (81,930) | 20.3 (7,138) | 18.4 (992) | 20.8 (62,145) | 20.0 (11,655) |
| 3rd | 18.7 (74,335) | 19.0 (6,687) | 19.1 (1,030) | 18.6 (55,676) | 18.8 (10,942) |
| 4th | 17.7 (70,423) | 19.2 (6,762) | 19.4 (1,043) | 17.3 (51,918) | 18.4 (10,700) |
| Highest | 17.6 (69,944) | 18.3 (6,413) | 21.7 (1,167) | 16.9 (50,557) | 20.3 (11,807) |
| Undefined | 1.4 (6,040) | 1.3 (455) | 5.2 (281) | 1.4 (4,213) | 1.9 (1,091) |
| **Residence location, %(n)** | | | | | |
| Urban | 91.0 (362,064) | 92.6 (32,533) | 85.5 (4,604) | 90.8 (271,789) | 91.4 (53,138) |
| Rural | 7.9 (31,556) | 6.7 (2,340) | 10.0 (537) | 8.2 (24,517) | 7.2 (4,162) |
| Missing | 1.1 (4,475) | 0.7 (268) | 4.5 (242) | 1.0 (3,096) | 1.4 (869) |
| **Zone, %(n)** | | | | | |
| Zone 1 (Urban) | 37.2 (148,162) | 61.7 (21,669) | 46.4 (2,496) | 31.1 (93,093) | 53.1 (30,904) |
| Zone 2 (Urban) | 40.0 (159,212) | 5.8 (2,029) | 31.4 (1,691) | 8.0 (23,808) | 31.4 (18,265) |
| Zone 3 (Rural) | 7.9 (31,337) | 24.1 (8,474) | 8.0 (431) | 43.7 (130,782) | 8.7 (5,069) |
| Zone 4 (Rural) | 8.6 (34,287) | 5.3 (1,848) | 6.1 (329) | 9.9 (29,557) | 4.4 (2,553) |
| Zone 5 (Rural) | 5.4 (21,629) | 2.6 (929) | 3.9 (212) | 6.6 (19,823) | 1.1 (665) |
| Missing | 0.9 (3,468) | 0.5 (192) | 4.2 (224) | 0.7 (2,339) | 1.3 (713) |
| **Patient attachment to a family physician******, %(n)** | | | | | |
| 75-100 | 26.9 (107,306) | 25.6 (8,978) | 28.0 (1,506) | 26.4 (79,069) | 30.5 (17,753) |
| 50 – <75 | 32.9 (130,985) | 32.9 (11,573) | 31.5 (1,697) | 32.7 (98,018) | 33.9 (19,697) |
| <50 | 37.4 (148,724) | 39.4 (13,828) | 32.2 (1,735) | 38.1 (114,155) | 32.7 (19,006) |
| Missing | 2.8 (11,080) | 2.1 (762) | 8.3 (445) | 2.8 (8,160) | 2.8 (1,713) |

*(Continued)*

| Patient Characteristics | | | | | |
|---|---|---|---|---|---|
| | Overall (n = 398,095) | Patients seeing salary-based specialist physicians (n = 40,524) | | Patients seeing FFS[+] specialist physicians (n = 357,571) | |
| | | Internal medicine n = 35,141 | Cardiologist n = 5,383 | Internal medicine = 299,402 | Cardiologist n = 58,169 |
| Patient age, mean (SD), years | 42.0 (13.7) | 39.4 (13.5) | 41.6 (15.2) | 41.7 (13.6) | 45.3 (13.7) |
| **Physician characteristics** | | | | | |
| | overall (n = 699) | Salary-based (n = 167) | | FFS (n = 532) | |
| | | Internal medicine n = 113 | Cardiologist n = 54 | Internal medicine n = 438 | Cardiologist n = 94 |
| Physician age, mean (SD), y | 43.5 (11.2) | 43.6 (9.2) | 47.1 (10.2) | 42.2 (11.7) | 47.0 (10.5) |
| **Physician gender, %(n)** | | | | | |
| Woman | 30.7 (215) | 48.7 (55) | 27.8 (15) | 30.6 (134) | 11.7 (11) |
| Man | 69.0 (482) | 51.3 (58) | 72.2 (39) | 69.0 (302) | 88.3 (83) |
| Missing | 0.3 (2) | 0.0 (0) | 0.0 (0 | 0.4 (2) | 0.0 (0) |
| **Country of training, %(n)** | | | | | |
| Canada | 65.5 (458) | 74.3 (84) | 79.6 (43) | 61.2 (268) | 67.0 (63) |
| High/upper middle income countries with similar medical training system to Canada ** | 8.7 (61) | 8.9 (10) | 7.4 (4) | 8.2 (36) | 11.7 (11) |
| Other high/upper middle-income country*** | 14.5 (101) | 8.0 (9) | 11.1 (6) | 15.8 (69) | 18.1 (17) |
| Lower middle/low-income country*** | 9.3 (65) | 8.0 (9) | 0.0 (0) | 12.6 (55) | 1.1 (1) |
| Unknown/ Missing | 2.0 (14) | 0.8 (1) | 1.9 (1) | 2.2 (10) | 2.1 (2) |
| **Clinical workload****, %(n)** | | | | | |
| < 15% | 27.3 (191) | 35.4 (40) | 31.5 (17) | 27.4 (120) | 14.9 (14) |
| 15-30% | 36.6 (256) | 51.3 (58) | 64.8 (35) | 29.0 (127) | 38.3 (36) |
| > 30% | 36.1 (252) | 13.3 (15) | 3.7 (2) | 43.6 (191) | 46.8 (44) |
| **Years in practice, %(n)** | | | | | |
| < 15 | 49.1 (343) | 43.4 (49) | 37.0 (20) | 55.5 (243) | 33.0 (31) |
| 15-30 | 33.6 (235) | 45.1 (51) | 38.9 (21) | 28.1 (123) | 42.6 (40) |
| > 30 | 15.6 (109) | 10.6 (12) | 22.2 (12) | 14.6 (64) | 22.3 (21) |
| Missing | 1.7 (12) | 0.9 (1) | 1.9 (1) | 1.8 (8) | 2.1 (2) |
| **Quartiles of patient volume*****, %(n)** | | | | | |
| 4 (highest) (>13 claims per day) | 24.3 (170) | 5.3 (6) | 0.0 (0) | 30.6 (134) | 31.9 (30) |
| 3 (10–13 claims per day) | 23.3 (163) | 17.7 (20) | 1.9 (1) | 26.7 (117) | 26.6 (25) |
| 2 (7–9 claims per day) | 25.6 (179) | 36.3 (41) | 20.4 (11) | 23.1 (101) | 27.7 (26) |
| 1 (lowest) (<7 claims per day) | 26.8 (187) | 40.7 (46) | 77.7 (42) | 19.6 (86) | 13.8 (13) |
| **Zone, %(n)** | | | | | |
| Zone 1 (Urban) | 43.8 (306) | 64.6 (73) | 61.1 (33) | 34.9 (153) | 50.0 (47) |
| Zone 2 (Urban) | 43.6 (305) | 35.4 (40) | 38.9 (21) | 46.4 (203) | 43.6 (41) |
| Zone 3 (Rural) | 4.7 (33) | 0.0 (0) | 0.0 (0) | 7.1 (31) | 2.1 (2) |
| Zone 4 (Rural) | 3.2 (22) | 0.0 (0) | 0.0 (0) | 4.8 (21) | 1.1 (1) |
| Zone 5 (Rural) | 4.4 (31) | 0.0 (0) | 0.0 (0) | 6.4 (28) | 3.2 (3) |
| Missing | 0.3 (2) | 0.0 (0) | 0.0 (0) | 0.4 (2) | 0.0 (0) |

+FFS = Fee-For-Service

* Generally, these were non-cardiac comorbidities including alcohol use disorder, asthma, lymphoma, metastatic cancer, nonmetastatic cancer, pain, chronic pulmonary disease, hepatitis B, cirrhosis, dementia, depression, epilepsy, hypertension, hyperthyroidism, inflammatory bowel disease, irritable bowel syndrome, multiple sclerosis, Parkinson disease, peptic ulcer disease, psoriasis, arthritis, schizophrenia, constipation. Patients with cardiac comorbidities were excluded from the cohort.

*(Continued)*

**Table 1.** (Continued)

\*\*Countries with residency programs considered to have equivalent training include United States, South Africa, UK, Australia, New Zealand, Ireland, Hong Kong, Singapore, and Switzerland.

\*\*\* Country income level was determined using World Bank Classification. Reference: https://datahelpdesk.worldbank.org/knowledgebase/articles/906519-world-bank-country-and-lending-groups

\*\*\*\* clinical workload defined as the average proportion each specialist works clinically each year [total number of days working clinically each years/\*365.25) \*100].

\*\*\*\*\* Patient volume: Patient volume is defined as the mean number of patient visits per day. Derived by dividing the total number of outpatient billing claims by the total number of outpatient billing days within the study period. Billing claims are restricted to the same consult codes used to define index visits. Billing days were days with two or more outpatient claims for clinic visits.

\*\*\*\*\*\*Primary care attachment (based on the concept of relational continuity) is defined as infrequent (1–2 visits), high (>75% of patients with three or more visits made to same physician), medium (50%–75% of patients with 3 or more visits to same physician), and low (<50% of visits made to any one primary care physician).

### Physician characteristics associated with frequent use of non-invasive cardiac imaging

For physicians, the odds of being a frequent user was significantly higher for men (OR=2.7, 95% CI: 1.8–4.0), those aged 40-61y (OR=1.6, 95% CI: 1.1–2.2), and foreign medical school graduates from high/upper middle-income countries (OR=1.9, 95% CI: 1.1–3.3) (Fig 4). Salary-based internal medicine specialists (OR=0.0043, 95% CI: 0.0014–0.014), FFS internal medicine specialists (OR=0.032, 95% CI: 0.015–0.067), and salary-based cardiologists (OR=0.13, 95% CI: 0.051–0.31) had lower odds of being a frequent user compared to FFS cardiologists.

### Sensitivity analyses

The results of the first four sensitivity analyses were consistent with the primary analysis (Table 2). We observed slightly more pronounced effect sizes in the sensitivity analysis using the broader cohort of patients seeing cardiologists and any internal medicine specialists (with no restriction to cardiology claims) and the sensitivity analysis where we restricted the cohort to index visits for anginal or anginal equivalent diagnostic codes for nuclear myocardial perfusion imaging and exercise treadmill testing.

In the last sensitivity analysis, the FFS cardiologists were further categorized into those who interpreted cardiac tests (FFS interpreter – n = 58) and those who did not (n = 36). Compared to FFS cardiologists who were not paid to interpret imaging tests, we identified ORs of 0.54 (95%CI 0.30–1.00), 0.11 (95%CI 0.064–0.19), 0.20 (95%CI 0.12–0.34) and 3.37 (95%CI 1.82–6.25) for salary-based cardiologists, salary-based internal medicine physicians, FFS internal medicine physicians, and FFS interpreters (Table 3).

### Discussion

In this population-based retrospective cohort study of specialist physicians in Alberta, one-quarter of patients at low cardiovascular risk received cardiac imaging overall, but nearly three-quarters of low-risk patients seeing FFS cardiologists received cardiac imaging. After controlling for patient- and zone-level differences, patients seen by salary-based internal medicine and cardiology physicians and FFS internal medicine specialists were significantly less likely to receive cardiac imaging compared to patients seeing FFS cardiologists. Frequent users of cardiac imaging were more likely to be men, and FFS cardiologists.

Compared with other research, we found that physician factors were more strongly associated with cardiac imaging rates than was geographic variation [14,21,22]. Like other studies, we found that variation in use of low-value testing is more strongly associated with physicians rather than patient factors [9,23,24].

Throughout, we have been cautious about implying that the cardiac imaging in this study was low value, though our definition would generally be consistent with the Choosing Wisely Canada recommendations against non-invasive cardiac

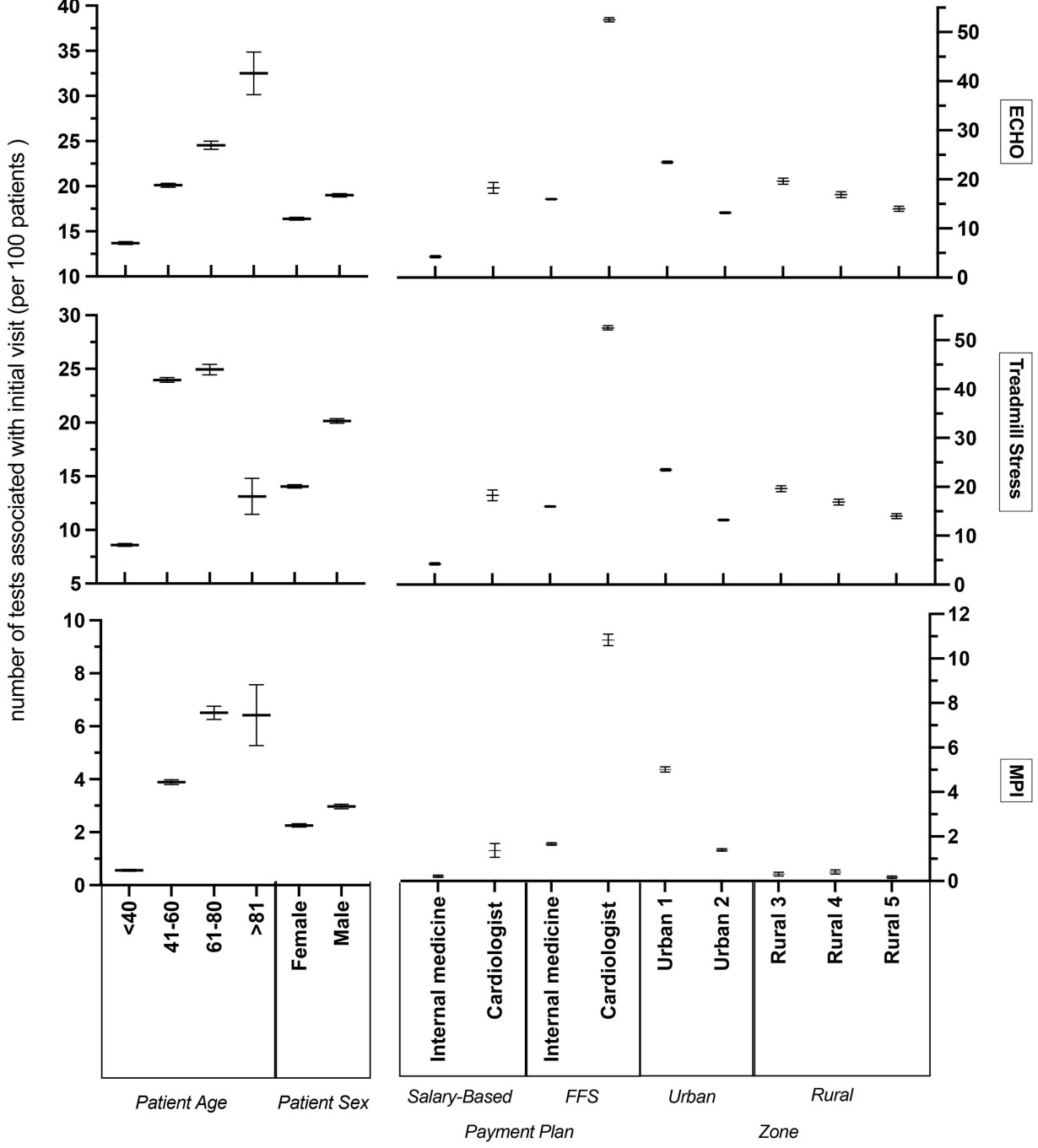

**Fig 2. Number of tests per 100 patients (overall and by test) by patient, physician, and geographic characteristics (with 95% confidence intervals).**

imaging in asymptomatic patients at low cardiovascular risk [25]. Importantly, we do not have information on patient symptoms or physical findings so it is possible that patients referred to FFS cardiologists (compared with internal medicine specialists or salaried cardiologists) had different presenting features or clinical findings that would be more likely

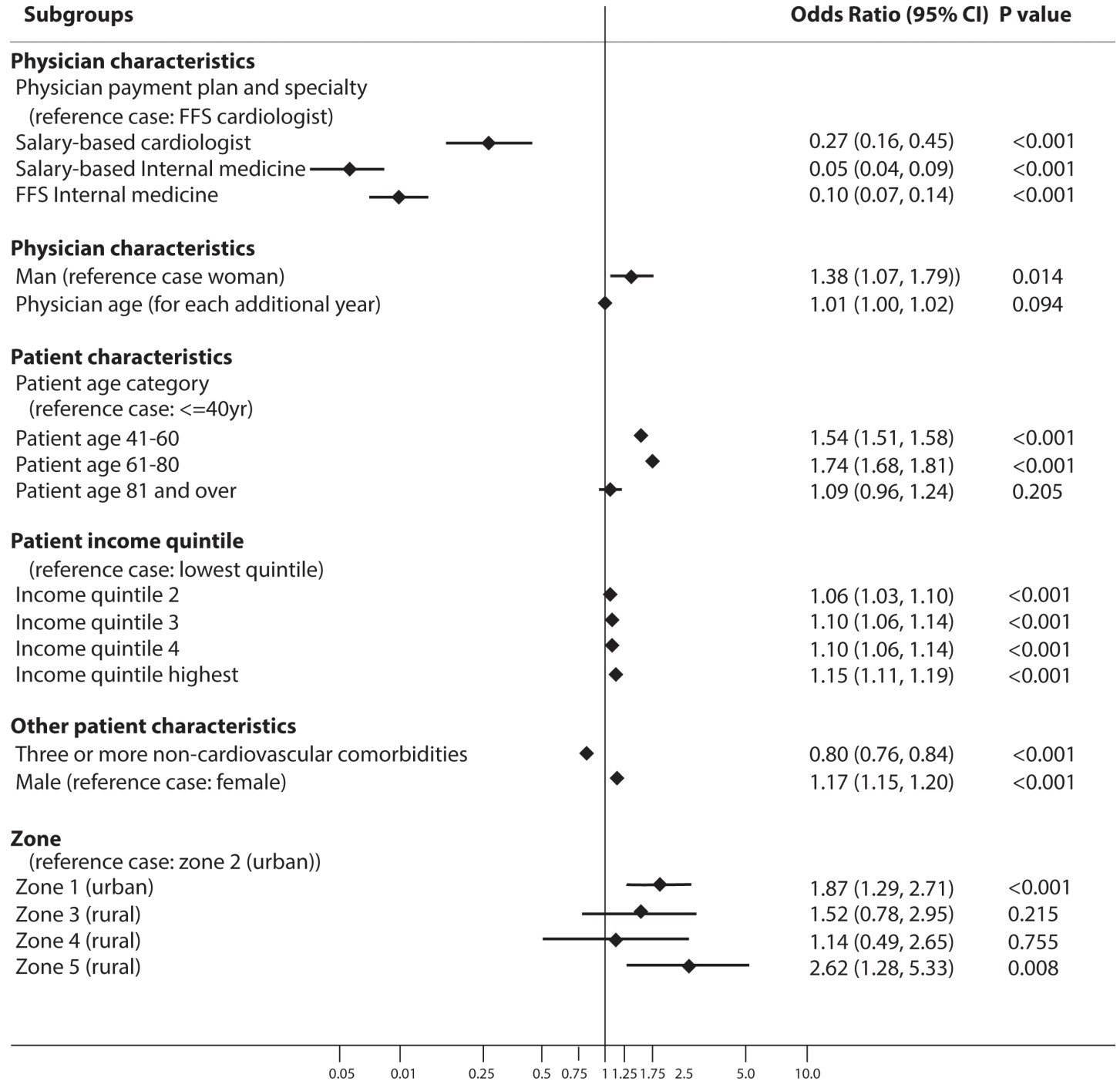

**Fig 3. Forest plot displaying multilevel, multivariable logistic regression model exploring the association between patient, physician, and geographic characteristics and cardiac testing for patients at low cardiovascular risk.**

to warrant testing but were not captured by administrative data. However, physician factors accounted for the majority of variation in cardiac testing and relationships between payment model and imaging intensity were consistent in each sensitivity analysis. Also, cardiac imaging was lower among patients seen by salaried cardiologists than by FFS cardiologists,

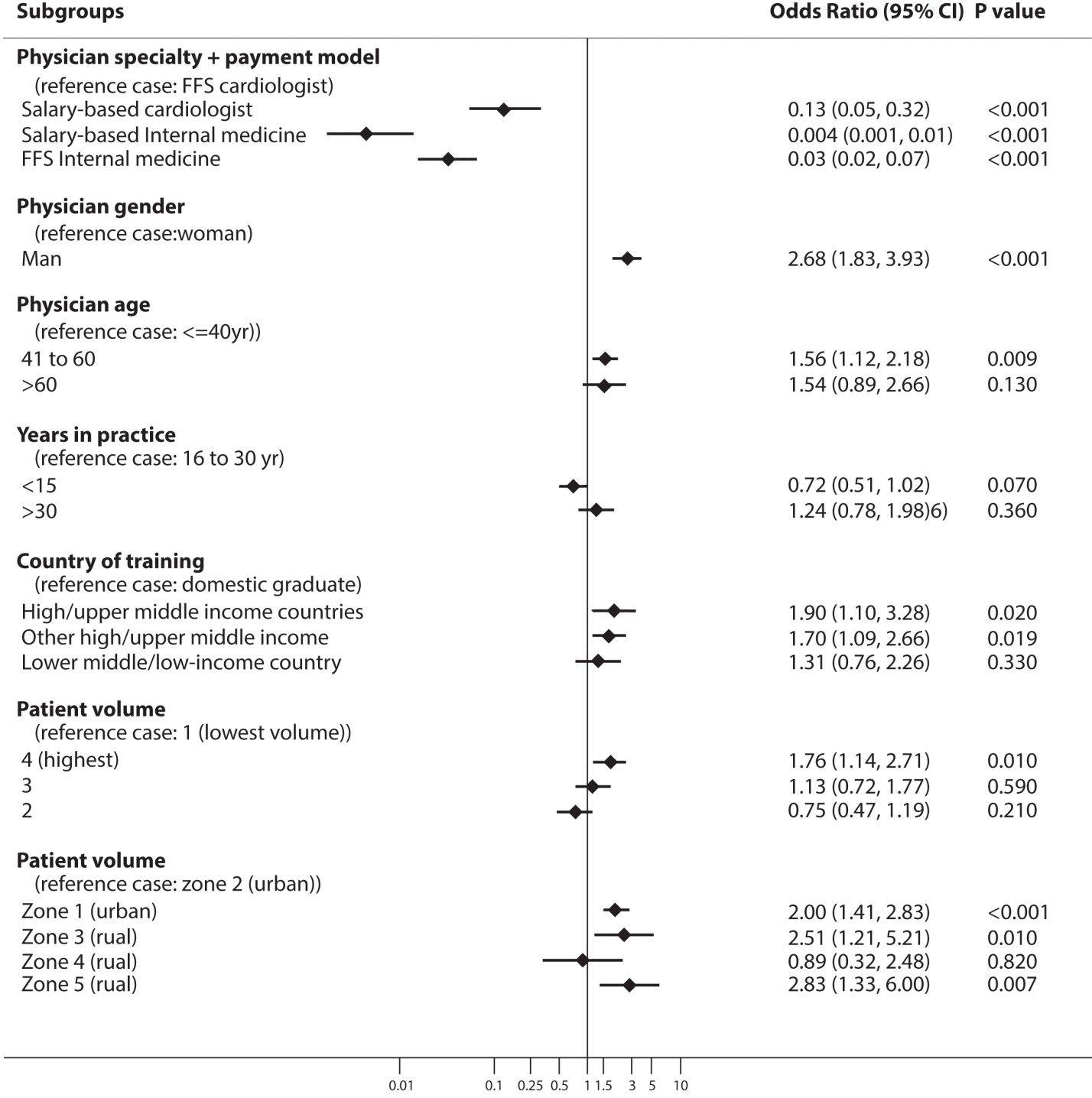

**Fig 4. Forest plot displaying the association between being a frequent user (a physician in the highest tertile of cardiac imaging use) and physician characteristics.**

**Table 2. Sensitivity analyses varying cohort inclusion criteria\*, each reporting the association between being a frequent user (a physician in the highest tertile of cardiac imaging use) and physician characteristics.**

| | Physician specialty and Payment model | | |
|---|---|---|---|
| | Salary-based Cardiologist (Reference group FFS Cardiologist) | Salary-based Internal medicine (Reference group FFS Cardiologist) | FFS Internal Medicine (Reference group FFS Cardiologist) |
| **Primary analysis:** | | | |
| Odds ratio | 0.13 | 0.004 | 0.032 |
| P-value | <0.001 | <0.001 | <0.001 |
| 95% confidence interval | 0.05–0.31 | 0.001–0.01 | 0.015–0.067 |
| **Broader cohort including all patients seeing cardiologists and internal medicine specialists (with no restriction to cardiology diagnostic claims)** | | | |
| Odds ratio | 0.19 | 0.008 | 0.029 |
| P-value | 0.001 | <0.001 | <0.001 |
| 95% confidence interval | 0.07–0.51 | 0.003–0.02 | 0.013–0.063 |
| **Baseline patient cohort but outcome limited to nuclear myocardial perfusion imaging, and exercise treadmill testing (excluded transthoracic echocardiogram testing)** | | | |
| Odds ratio | 0.14 | 0.02 | 0.15 |
| P-value | 0.002 | <0.001 | <0.001 |
| 95% confidence interval | 0.04–0.48 | 0.003–0.16 | 0.086–0.28 |
| **Cohort restricted to index visits for anginal or anginal equivalent diagnostic codes (i.e., ICD9 codes 785, 786) and outcome limited to nuclear myocardial perfusion imaging, and exercise treadmill testing** | | | |
| Odds ratio | 0.22 | 0.06 | 0.13 |
| P-value | <0.001 | <0.001 | <0.001 |
| 95% confidence interval | 0.10–0.45 | 0.03–0.13 | 0.081–0.21 |
| **Ultra-low cardiovascular risk cohort: Cohort excluded patients that developed cardiovascular disease (MI, CHF, stroke, PVD, CABG/PCI, IHD, valvular heart disease) or arrythmia or required pacemaker/ICDs/CRTs in the subsequent year after index consult, implying that the excluded patients had symptoms concerning for cardiac disease that warranted testing** | | | |
| Odds ratio | 0.14 | 0.004 | 0.03 |
| P-value | <0.001 | <0.001 | <0.001 |
| 95% confidence interval | 0.05–0.34 | 0.001–0.014 | 0.02–0.07 |

\*For simplicity, only results of the physician payment/specialty variable are reported in this table, but all analyses control for the covariates listed in Fig 4

**Table 3. Sensitivity analysis including an additional category of FFS interpreter, repeating the baseline analysis and reporting the association between patient, physician, and geographic characteristics and cardiac testing for patients at low cardiovascular risk.**

| Physician payment plan and specialty Reference case: FFS cardiologist | Odds ratio | p-value | 95% confidence interval | |
|---|---|---|---|---|
| Salary-based cardiologist | 0.54 | 0.052 | 0.30 | 1.00 |
| Salary-based internal medicine | 0.11 | <0.001 | 0.064 | 0.19 |
| FFS internal medicine | 0.20 | <0.001 | 0.12 | 0.34 |
| FFS interpreter | 3.37 | <0.001 | 1.82 | 6.25 |

\*For simplicity, only results of the physician payment/specialty variable are reported in this table, but all analyses control for the covariates listed in Fig 3

suggesting that payment model is a more important determinant of test utilization than clinical specialty. These findings raise the question of whether all the cardiac imaging was appropriate, a concern since false positive testing (more common in patients at low cardiac risk) may lead to overdiagnosis and further invasive testing or treatment, with negative consequences for patients and cost implications for the healthcare system [26–28].

In Alberta, private testing facilities expanded significantly in the 1990s in response to concerns about access to non-invasive cardiac testing. The number of private testing facilities in the province has increased substantially over time, particularly in one of Alberta's larger cities, which now has relatively short wait times, indicating ample testing supply. We noted that use of cardiac imaging was highest in this urban center despite controlling for differences in patients and physicians, suggesting unwarranted variation. In addition, our sensitivity analysis showed that patients seeing FFS cardiologists who also bill to interpret cardiac tests have substantially higher odds of receiving cardiac imaging compared to other FFS cardiologists. Although we do not have data on who owns private imaging facilities, FFS cardiologists that are paid for interpretation of imaging studies [1] are more likely to have an equity interest in imaging facilities, highlighting the potential trade-off between improved patient access and over-testing – and its impact on patients.

Our results provide empirical evidence for supplier-induced demand, since physicians who can also bill for cardiac imaging have a direct financial benefit to order more tests. The financial incentives for FFS physicians who do not have a direct financial stake in cardiac testing are more indirect. Ordering a test is often faster than explaining why additional testing is not required, and physicians paid FFS can earn more through shorter visits. Beyond financial incentives, there are other causes of variation in care, indicating the need to develop focused interventions aimed at improving test ordering in patients at low cardiovascular risk. Examples could include academic detailing (one-on-one, evidence-based educational outreach to clinicians) or focused awareness/education campaigns rooted in non-judgmental communication, use of Electronic Health Record-integrated clinical decision tools, provision of concise peer reporting and comparisons, and leveraging the use of clinical champions [29–31]. Changes at the system level may also be useful, including regulatory changes (e.g., restricting the ability for self-referrals and ensuring that ownership does not impact practice patterns, as is done with physicians who own pharmacies), implementing new alternate payment models and/or central triage for cardiac imaging to increase appropriateness of testing.

Although our findings show that physician factors are more important than patient factors in determining the use of cardiac imaging in low-risk patients, some imaging tests may occur as a result of patient demand and/or referring primary care physician expectations. While more research is required to understand patient demand for potentially low-value cardiac imaging, multiple studies have shown that better patient engagement can reduce low-value tests. For example, with patient education, the use of spinal imaging was reduced by 30% and shared decision-making support reduced preference sensitive surgeries by 10% [29,30].

## Limitations

Our analysis has limitations that should be considered. We used administrative data, which lacks specific information on clinical symptoms and physical findings which may influence testing decisions. We did not have data to indicate which physician ordered the imaging studies, but our consultation with specialists confirmed that some specialists routinely ordered cardiac imaging prior to patients' initial visits and others ordered them after the visits. Given that there is a short wait time (<30 days) for cardiology imaging studies in Alberta, but often long waits to have a consultation with a specialist, we assumed that imaging ordered within the 30 days before or 90 days after were related to the clinic visit. Any misclassification resulting from imaging studies being ordered by primary care physicians rather than specialists should have biased toward the null and so is unlikely to explain the observed associations.

Future studies should assess whether our findings extend to other jurisdictions, measure and control for why the test was ordered, and assess the optimal strategies to reduce use of low-value cardiac testing.

## Conclusions

How physicians are paid, particularly in systems with for-profit cardiac testing and physician ownership, is associated with the use of cardiac imaging in patients at low cardiovascular risk. This has implications for health systems aiming to reduce low-value care, and is an important lesson for jurisdictions considering implementing or expanding private for-profit cardiac testing facilities. For jurisdictions with private for-profit diagnostic imaging facilities (particularly those with substantial physician ownership), physician payment reforms, restrictions on self-referral to testing, and active strategies to implement Choosing Wisely recommendations offer an opportunity to intervene at the system and physician level to improve the appropriateness and efficiency of cardiac care.

## Supporting information

**S1 Table. List of consultation codes used to define physician index visits.**
(PDF)

**S2 Table. List of ICD9 codes used to define cardiovascular disease and cardiovascular risk factors.**
(PDF)

**S3 Table. Procedure codes used to define cardiac testing.**
(PDF)

**S4. Demographic characteristics of people who received and did not receive cardiac imaging.**
(PDF)

**S5 Table. Explained and unexplained physician, patient, and geographic-level variation across models estimating odds of cardiac testing.**
(PDF)

**S6 Table. Highlighting variation in likelihood of cardiac testing across different physician groups.**
(PDF)

## Acknowledgments

**Disclaimer:** This study is based in part on data provided by Alberta Health and Alberta Health Services. The interpretation and conclusions contained herein are those of the researchers and do not necessarily represent the views of the Government of Alberta or Alberta Health Services. Neither the Government of Alberta nor, Alberta Health or Alberta Health Services express any opinion in relation to this study.

## Author contributions

**Conceptualization:** Yewande Kofoworola Ogundeji, Derek S. Chew, Braden J. Manns.

**Data curation:** Braden J. Manns.

**Formal analysis:** Yewande Kofoworola Ogundeji, Flora Au.

**Funding acquisition:** Yewande Kofoworola Ogundeji, Braden J. Manns.

**Investigation:** Yewande Kofoworola Ogundeji, Amity E. Quinn, Flora Au, Braden J. Manns.

**Methodology:** Yewande Kofoworola Ogundeji, Amity E. Quinn, Derek S. Chew, Stephen B. Wilton, Braden J. Manns.

**Project administration:** Yewande Kofoworola Ogundeji.

**Resources:** Braden J. Manns.

**Supervision:** Braden J. Manns.

**Validation:** Yewande Kofoworola Ogundeji, Amity E. Quinn, Derek S. Chew, Stephen B. Wilton, Matthew T. James, Marcello Tonelli, Braden J. Manns.

**Visualization:** Yewande Kofoworola Ogundeji, Amity E. Quinn, Flora Au, Braden J. Manns.

**Writing – original draft:** Yewande Kofoworola Ogundeji, Braden J. Manns.

**Writing – review & editing:** Yewande Kofoworola Ogundeji, Amity E. Quinn, Derek S. Chew, Stephen B. Wilton, Matthew T. James, Marcello Tonelli, Braden J. Manns.

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
