## [Decision Letter · Decision Letter 0]

10 Jun 2025

Dear Dr. Manns,

Thank you for submitting your manuscript to PLOS ONE. After careful consideration, we feel that it has merit but does not fully meet PLOS ONE’s publication criteria as it currently stands. Therefore, we invite you to submit a revised version of the manuscript that addresses the points raised during the review process.

We look forward to receiving your revised manuscript.

Kind regards,

Jialing Lin

Academic Editor

PLOS ONE

Journal Requirements:

4. In the online submission form, you indicated that we cannot make our dataset available to other researchers due to our contractual arrangements with the provincial health ministry (Alberta Health), who is the data custodian. Researchers may make requests to obtain a similar dataset at https://absporu.ca/research-services/service-application/.

A Canadian Institutes of Health Research Foundation Grant to Manns and a Data Collection Grant from the Clinical Research Fund, University of Calgary to Ogundeji and Manns funded this study

6. Please remove all personal information, ensure that the data shared are in accordance with participant consent, and re-upload a fully anonymized data set.

Reviewers' comments:

Reviewer's Responses to Questions

**Comments to the Author**

1. Is the manuscript technically sound, and do the data support the conclusions?

Reviewer #1: Yes

Reviewer #2: Partly

2. Has the statistical analysis been performed appropriately and rigorously?

Reviewer #1: Yes

Reviewer #2: Yes

3. Have the authors made all data underlying the findings in their manuscript fully available?

Reviewer #1: Yes

Reviewer #2: Yes

4. Is the manuscript presented in an intelligible fashion and written in standard English?

Reviewer #1: Yes

Reviewer #2: Yes

Reviewer #1: This paper is well-written, clearly structured, and addresses an important issue in health services research. While the study provides valuable insights, the following suggestions are intended to further strengthen its quality and enhance its potential for publication.

• The study addresses a relevant and timely topic in health services research by examining how physician payment models influence cardiac imaging in low-risk populations.

• The use of a large retrospective cohort from administrative health data in Alberta enhances the robustness and generalizability of the findings.

• The manuscript is generally well-written, with clear articulation of the research objective, methods, results, and conclusions.

• The study design is appropriate for the research question, and the use of multilevel, multivariable logistic regression is methodologically sound.

• The classification of patients into low cardiovascular risk is appropriately done using administrative data, although the limitations of such classification should be more clearly acknowledged.

• The description of the payment models (fee-for-service vs. salary-based) could benefit from additional context to aid readers unfamiliar with the Canadian healthcare system.

• The finding that salary-based physicians, particularly internal medicine specialists, are less likely to order imaging is compelling and aligns with hypotheses regarding financial incentives.

• The discussion provides a good synthesis of the results, although more emphasis could be placed on potential policy implications of these findings.

• The conclusion is supported by the presented data and underscores the role of physician incentives in potentially unnecessary care.

• The authors performed several sensitivity analyses to confirm the robustness of their findings, which is commendable.

• The manuscript could improve by including a brief explanation of the ethical approval process, particularly given the use of administrative health data.

• Some figures or tables (if available in the full version) may need more detailed legends to stand alone for clarity.

• There is no indication of ethical concerns or misconduct related to research design, data analysis, or authorship.

Reviewer #2: Good work,the topic is timely and relevant, addressing an underexplored area within health services research. However, the manuscript would benefit from several improvements to maximize its global relevance, contextual clarity, and analytical completeness. The comments below are intended to guide your revision constructively.

1. Title and Keywords

1.1 The title is technically accurate but long and clinical. Consider simplifying it to improve accessibility, e.g., Physician Payment Models and Cardiac Imaging in Low-Risk Patients: A Population-Based Cohort Study in Alberta, Canada.

1.2 The keywords do not sufficiently capture the paper’s scope or uniqueness. Suggested replacements: physician remuneration, cardiac test overuse, primary care incentives, Alberta health system, low-value care.

2. Abstract

2.1 Avoid jargon like “receipt of imaging” — use more universally understood terms.

2.2 The conclusion is too narrowly framed. Consider broadening its relevance to readers unfamiliar with Canadian payment models.

3. Introduction

3.1 The introduction is well-focused on Choosing Wisely and overuse but lacks global context. A comparison of physician payment models across countries (e.g., FFS, capitation, salary) would help orient an international audience.

3.2 The literature cited is mainly Canadian; expanding to include studies from the US, UK, or Australia would strengthen framing. broaden this

3.3 The study objective is clearly stated but could be linked more explicitly to the broader issue of health system design and policy reform.

4. Methods

4.1 The study design is appropriate, but the term “retrospective cohort” may overstate longitudinal rigor. A more precise description would be: “A population-based cohort study using health service data in Alberta, Canada.”

4.2 The assumption of imaging attribution to the nearest outpatient visit should be explained and justified.

4.3 The impact of COVID-19 on test volumes in 2020 is not discussed. A sensitivity analysis excluding 2020 or a clear rationale for inclusion is needed.

4.4 While physician- and patient-level variables are handled well, missing clinical variables (e.g., symptoms) should be flagged more explicitly in Methods, not just in Discussion.

4.5 Ethics approval is mentioned but would benefit from more clarity — include board name, reference number, or waiver status.

5. Results

5.1 Results are presented clearly but are overly text-heavy. Consider including:

A forest plot of ORs

A STROBE-style flow diagram for cohort construction

A bar or line chart showing temporal trends (2012–2020)

5.2 The interpretation of ORs should be supported with absolute risk or test rate examples to enhance real-world understanding.

5.3 No subgroup results by sex, rurality, or age are discussed — such insights could enrich interpretation.

5.4 Specialty comparisons could be disaggregated further for clarity (e.g., GP vs internist vs cardiologist).

5.5 A map or graph showing regional imaging variation would be informative.

6. Discussion

6.1 The discussion is balanced but too narrowly scoped.

6.2 Future research directions are not addressed. Consider suggesting:

Multi-jurisdictional comparisons

Studies with clinical data to assess appropriate vs inappropriate imaging

Implications for payment reform or low-value care monitoring

6.3 The conclusion line is overly technical: “Physician payment models and specialty are strongly associated with receipt of imaging…” Consider rephrasing for global readability, e.g.:

“How physicians are paid appears to influence imaging use patterns, even in low-risk patients. This has implications for health systems aiming to reduce low-value care.”

7. Global Relevance and Framing

7.1 The framing leans heavily on Alberta/Canada context. Consider:

Generalizing findings into broader health economics or system design language

Citing international frameworks (e.g., OECD health data, Choosing Wisely Global)

7.2 While local details are critical, striking a balance between specificity and transferability would increase the paper’s reach and relevance.

This manuscript is well-conceived and methodologically sound but would benefit from revisions that strengthen its clarity, expand its global framing, and deepen result interpretability. Addressing the issues outlined above will improve the paper . all the best

**Do you want your identity to be public for this peer review?** For information about this choice, including consent withdrawal, please see our Privacy Policy

Reviewer #1: **Yes: ** Mohsin Hassan Alvi

Reviewer #2: **Yes: ** Thamburaj Anthuvan

---

## [Author Response · Author response to Decision Letter 1]

22 Sep 2025

Journal Requirements:

Thank you. We have done so.

Thank you. It appears that I had multiple records in PLOS but that has been corrected, and my ORCID number now appears.

See below

4. In the online submission form, you indicated that we cannot make our dataset available to other researchers due to our contractual arrangements with the provincial health ministry (Alberta Health), who is the data custodian. Researchers may make requests to obtain a similar dataset at https://absporu.ca/research-services/service-application/.

I corresponded with Dr Lin on this matter on June 24 (response from Ethan Krajewski, Peer Review Operations Specialist, suggesting our response was satisfactory). I will reiterate that here.

“There are legal restrictions on sharing our dataset, even fully de-identified data, from our provincial health ministry, who is the formal custodian for the health administrative data that we use for this study. We have been advocating for a change to their policy, which is embedded within the contract through which we access the health data, but have not been successful. Attached, please see a letter from the Alberta government health ministry confirming this legal restriction, and that we would be in breach of the law if we were to make a de-identified dataset available.

As an alternative, those interested in conducting similar analyses can make requests to obtain a similar dataset at https://absporu.ca/research-services/service-application/ ."

As you note above, we have informed readers of this opportunity.

Unfortunately, this policy doesn’t just apply to this one study – it applies to all research conducted by our group using this dataset. Of note, this rationale was recently accepted by PLOSOne within the past year - https://pmc.ncbi.nlm.nih.gov/articles/PMC11156280/ using the same Data Sharing statement. Thank you very much for considering this exemption.

A Canadian Institutes of Health Research Foundation Grant to Manns and a Data Collection Grant from the Clinical Research Fund, University of Calgary to Ogundeji and Manns funded this study

Thank you. I can confirm that “The funders had no role in study design, data collection and analysis, decision to publish, or preparation of the manuscript." This has now been added.

6. Please remove all personal information, ensure that the data shared are in accordance with participant consent, and re-upload a fully anonymized data set.

See response above. Unfortunately, we are not able to share our de-identified data.

See response above. Unfortunately, we are not able to share our de-identified data.

Reviewers' comments:

Reviewer's Responses to Questions

Comments to the Author

1. Is the manuscript technically sound, and do the data support the conclusions?

Reviewer #1: Yes

Reviewer #2: Partly

2. Has the statistical analysis been performed appropriately and rigorously?

Reviewer #1: Yes

Reviewer #2: Yes

3. Have the authors made all data underlying the findings in their manuscript fully available?

Reviewer #1: Yes

Reviewer #2: Yes

4. Is the manuscript presented in an intelligible fashion and written in standard English?

Reviewer #1: Yes

Reviewer #2: Yes

Thank you

5. Review Comments to the Author

Reviewer #1: This paper is well-written, clearly structured, and addresses an important issue in health services research. While the study provides valuable insights, the following suggestions are intended to further strengthen its quality and enhance its potential for publication.

• The study addresses a relevant and timely topic in health services research by examining how physician payment models influence cardiac imaging in low-risk populations.

• The use of a large retrospective cohort from administrative health data in Alberta enhances the robustness and generalizability of the findings.

• The manuscript is generally well-written, with clear articulation of the research objective, methods, results, and conclusions.

• The study design is appropriate for the research question, and the use of multilevel, multivariable logistic regression is methodologically sound.

• The classification of patients into low cardiovascular risk is appropriately done using administrative data, although the limitations of such classification should be more clearly acknowledged.

• The description of the payment models (fee-for-service vs. salary-based) could benefit from additional context to aid readers unfamiliar with the Canadian healthcare system.

• The finding that salary-based physicians, particularly internal medicine specialists, are less likely to order imaging is compelling and aligns with hypotheses regarding financial incentives.

• The discussion provides a good synthesis of the results, although more emphasis could be placed on potential policy implications of these findings.

• The conclusion is supported by the presented data and underscores the role of physician incentives in potentially unnecessary care.

• The authors performed several sensitivity analyses to confirm the robustness of their findings, which is commendable.

• The manuscript could improve by including a brief explanation of the ethical approval process, particularly given the use of administrative health data.

Thank you. We have provided additional detail.

• Some figures or tables (if available in the full version) may need more detailed legends to stand alone for clarity.

Thank you. We have modified several of the tables / figures based on reviewers’ comments and added additional detail in the legends

• There is no indication of ethical concerns or misconduct related to research design, data analysis, or authorship.

Reviewer #2: Good work,the topic is timely and relevant, addressing an underexplored area within health services research. However, the manuscript would benefit from several improvements to maximize its global relevance, contextual clarity, and analytical completeness. The comments below are intended to guide your revision constructively.

1. Title and Keywords

1.1 The title is technically accurate but long and clinical. Consider simplifying it to improve accessibility, e.g., Physician Payment Models and Cardiac Imaging in Low-Risk Patients: A Population-Based Cohort Study in Alberta, Canada.

Thank you. We have accepted your suggestion (with one small wording change)

1.2 The keywords do not sufficiently capture the paper’s scope or uniqueness. Suggested replacements: physician remuneration, cardiac test overuse, primary care incentives, Alberta health system, low-value care.

Thank you. We have added physician remuneration, cardiac test overuse, low value care.

2. Abstract

2.1 Avoid jargon like “receipt of imaging” — use more universally understood terms.

Thank you. This has been changed to cardiac imaging throughout the manuscript.

2.2 The conclusion is too narrowly framed. Consider broadening its relevance to readers unfamiliar with Canadian payment models.

Thank you. We have broadened the conclusion, while keeping in mind that many PLOS ONE readers will be international

3. Introduction

3.1 The introduction is well-focused on Choosing Wisely and overuse but lacks global context. A comparison of physician payment models across countries (e.g., FFS, capitation, salary) would help orient an international audience.

An additional paragraph has been included in the introduction to include relevant literature across broader health systems context (see pages 3 and 4).

3.2 The literature cited is mainly Canadian; expanding to include studies from the US, UK, or Australia would strengthen framing. broaden this

An additional paragraph has been included in the introduction to include relevant literature across broader health system contexts (see pages 3 and 4). We’ve also augmented the evidence cited to include other countries.

3.3 The study objective is clearly stated but could be linked more explicitly to the broader issue of health system design and policy reform.

In addition to linking the study to global contexts, we’ve included additional text on the potential influence/impact on physician payment policy reforms

4. Methods

4.1 The study design is appropriate, but the term “retrospective cohort” may overstate longitudinal rigor. A more precise description would be: “A population-based cohort study using health service data in Alberta, Canada.”

The title and description have been changed to reflect this.

4.2 The assumption of imaging attribution to the nearest outpatient visit should be explained and justified.

Thank you. We are happy to expand, and have added some additional text to the limitation section of the paper. Without clinical data that indicated the ordering physician, we examined the distribution of imaging tests 90 days before patients’ initial visits and up to a year following their initial visits (see figure below). We determined that the majority of tests were ordered during the -30 to +90 day window (Figure). We also spoke to cardiologists about their test ordering practices. Some cardiology specialists routinely ordered testing in advance of their new visits while others ordered them afterwards. Given that there is a short wait time (<30 days) for cardiology imaging studies in Alberta, but often long waits to have a consultation with a specialist, we made the assumption that tests ordered within the 30 days before or 90 days after were related to the clinic visit.

We cannot be certain that the tests were ordered by the internist or cardiologist, but it seems unlikely that primary care physicians would change their own imaging ordering practices based on the payment model of the specialist they are referring patients to. So if primary care physicians were ordering many of the tests, we would expect this to dilute the observed association.

4.3 The impact of COVID-19 on test volumes in 2020 is not discussed. A sensitivity analysis excluding 2020 or a clear rationale for inclusion is needed.

Our study period did not extend into the period of the pandemic.

4.4 While physician- and patient-level variables are handled well, missing clinical v

---

## [Editor Report · Decision Letter 1]

26 Oct 2025

Physician payment models and cardiac imaging in patients at low cardiovascular risk: a population-based cohort study in Alberta, Canada.

PONE-D-25-18807R1

Dear Dr. Manns,

We’re pleased to inform you that your manuscript has been judged scientifically suitable for publication and will be formally accepted for publication once it meets all outstanding technical requirements.

Kind regards,

Jialing Lin

Academic Editor

PLOS ONE
---

## [Editor Report · Acceptance letter]

PONE-D-25-18807R1

PLOS ONE

Dear Dr. Manns,

I'm pleased to inform you that your manuscript has been deemed suitable for publication in PLOS ONE. Congratulations! Your manuscript is now being handed over to our production team.

Kind regards,

on behalf of

Dr. Jialing Lin

Academic Editor

PLOS ONE